# The frequency of complications in a cohort of patients diagnosed with hemophilia A and hemophilia B receiving prophylactic treatment in Colombia: A retrospective noninterventional study

Jorge E. Machado Alba[1]*, Juan David Wilches-Gutierrez[2], Diana Rocio Arias-Osorio[2], Juan Manuel Reyes[3], Maria Lourdes Nakandakari[3], Harrison David Ospina-Arzuaga[1], Andres Gaviria-Mendoza[1,4], Natalia Castaño-Gamboa[3], Luis Fernando Valladales-Restrepo[1,4], Manuel E. Machado-Duque[1,4]

1 Grupo de Investigación en Farmacoepidemiología y Farmacovigilancia, Universidad Tecnológica de Pereira-Audifarma S.A, Pereira, Colombia, 2 IPS Especializada, Bogotá, Colombia, 3 Pfizer Colombia, Bogotá, Colombia, 4 Grupo de Investigación en Biomedicina, Fundación Universitaria Autónoma de las Américas, Pereira, Colombia

* machado@utp.edu.co

## Abstract

### Introduction

Hemophilia A and B are disorders associated with the deficit of coagulation factors VIII and IX.

### Objective

Was to determine the incidence of complications in a cohort of patients diagnosed with moderate and severe hemophilia A or B under treatment in a specialized institution.

### Methods

A retrospective study of a cohort of patients with replacement therapy for hemophilia A or B, evaluating treatment and complications between January/2012 and July/2019. Sociodemographic, clinical and disease management-related variables were extracted from the medical records. Time to inhibitor development and rate associated with bleeding and hospitalizations were evaluated.

### Results

A total of 159 male patients were identified with hemophilia A (n = 140; 88.1%) and B (n = 19; 11.9%) with a mean follow-up of 5.9±2.3 years. The mean age was 23.6±16.1 years, hemophilia was reported as severe in 125 patients in hemophilia A (89.3%) and 13 patients in hemophilia B (68.4%). Primary prophylaxis was registered in 17.0% of patients, 44.7% secondary, and 38.3% tertiary, with recombinant factors (n = 84; 52.8%) followed by plasma

**Data Availability Statement:** Data availability: protocols.io Data access: dx.doi.org/10.17504/protocols.io.b4hwqt7e.

**Funding:** YES: This study was funded by Pfizer Colombia B8981001. The funder provided support in the form of salaries for authors Pfizer Colombia but did not have any additional role in the study design, data collection and analysis, decision to publish, or preparation of the manuscript. The specific roles of these authors are articulated in the 'author contributions' section.

**Competing interests:** yes: Reyes JM, Lourdes M and Castano N are paid employees of Pfizer Colombia. Juan David Wilches-Gutierrez, Diana Rocio Arias-Osorio are paid employees of IPS-Especializada. Andres Gaviria-Mendoza, Natalia Castaño-Gamboa, Luis Fernando Valladales-Restrepo, Manuel E. Machado-Duque and Jorge Machado-Alba are paid employees of Audifarma SA. Harrison David Ospina-Arzuaga do not have any conflict of interest. This does not alter our adherence to PLOS ONE policies on sharing data and materials.

derived factors (n = 75; 47.2%). The incidence of inhibitor development was 0.3 per 100 patients/year, with mean time to event of 509 days. The incidence of bleeding was 192 per 100 patients/year, especially at the joint (n = 99; 62.3%) and muscle (n = 25; 15.7%) level. The incidence of hospitalization was 3.7 per 100 patients/year.

## Conclusions

The most common complication was joint bleeding which was expected in this type of patients. Low proportion of patients developed factor inhibitors during the follow up.

## Introduction

Hemophilia A and hemophilia B are coagulation disorders related to the deficiency of factors VIII and IX, respectively. Hemophilia A and B are rare diseases worldwide, since their prevalence is approximately 1 case per 5,000 people [1, 2]. Patients with severe hemophilia are those with less than 1 IU/dL of the deficient factor [3, 4]; patients with levels between 1 and 5 IU/dL and between 5 IU/dL and 40 IU/dL are classified as moderate and mild, respectively [4]. The most severe forms of hemophilia are characterized by spontaneous bleeding in the joints and muscles [3]. Repeated bleeding in the same locations can lead to hemophilic arthropathy [5].

Quality care for these conditions, especially severe cases, requires the intervention of a multidisciplinary team focused on the prevention of bleeding and joint damage, the management of bleeding episodes, rehabilitation, pain management, psychosocial intervention and education [4, 6]. Through such efforts, joint damage is prevented or delayed, the patient's quality of life is preserved [7], and risk factors that negatively modify the evolution of the condition are mitigated [4]. In severe cases, prophylaxis serves as a pharmacological strategy focused on the prevention of bleeding, which can be primary, secondary or tertiary depending on the history of bleeding and joint damage [4, 8].

The complications related to treatment with clotting factor replacement that has the greatest impact is the development of inhibitors, since they affect the effectiveness of coagulation factors and increase the risks of morbidity and mortality and the costs to the health system [3, 4, 9, 10]. Real-world studies allow the evaluation of the clinical outcomes of patients under routine conditions. These findings provide decision-makers with information related to the effectiveness and safety of pharmacological technologies and consider scenarios that cannot always be evaluated in clinical studies. Additionally, they can complement available information to provide data on how doctors and patients choose treatments, dosages and regimens and on the frequency of use of health services and technologies and their effectiveness [5, 11].

Currently, in Colombia, there are no studies regarding the follow-up, treatment and outcomes of patients with hemophilia A and B; therefore, it is important to expand knowledge regarding this population. This need has led to the objective of evaluating the incidence of adverse outcomes in a cohort of patients diagnosed with hemophilia A and hemophilia B at a health care institution (Institución Prestadora de Servicios Especializada: IPS-E) of Audifarma SA) in Colombia. The IPS-E is a health institution dedicated to the care of patients affiliated with the Health System with high-cost diseases such as hemophilia, rheumatoid arthritis, among others, providing transdisciplinary care, the safe administration of health technologies, including medicines, aimed at restoring health and improving the quality of life.

## Materials and methods

This was a retrospective study of a dynamic cohort of patients diagnosed with hemophilia A and hemophilia B who entered the Coagulopathies Program of IPS-E for replacement therapy between January 2012 and July 2019. Given the descriptive nature of the study, the evolution of the treatment and the patients' clinical results were evaluated during the follow-up period. The index date was defined as the date of admission to the IPS-E Coagulopathies Program and the initiation of some recombinant or plasma-derived clotting factor replacement within a prophylaxis regimen. The follow-up time began from the index date to inhibitors development, ≥0.6 Bethesda units/milliliters (BU/ml), death, loss to follow-up, abandonment of the Program or July 2019, whichever came first.

Inclusion criteria were male patients diagnosed with hemophilia A or B, who had been registered with the Specialized Coagulopathies Program of IPS-E for 6 or more months and were undergoing prophylactic treatment with recombinant or plasma-derived factors. Those who presented inhibitors upon entering the program or who were participating in a clinical trial were excluded. Sample size estimation was not performed due to the design of the study; therefore, all of the patients in the Program who met the inclusion criteria were included in the cohort.

All clinical, treatment and outcome information were obtained from the patients' medical records, which were reviewed by two trained physicians, and all information was recorded in an electronic collection format. The following groups of variables were collected:

1. *Sociodemographic*: age, sex, city of residence, weight, height, family history of hemophilia and development of inhibitors.

2. *Clinical*: type of hemophilia, date of diagnosis, presence of arthropathy, history of bleeding and previous treatment.

3. *Pharmacological*: type of prophylaxis (primary, secondary, tertiary), type of factor, dose, administration interval, switches in coagulation factors, interruptions, and their causes.

4. *Adverse events*: bleeding, site of bleeding, trauma, surgeries, hospitalizations, development of inhibitors (BU), anaphylactic reactions or nephrotic syndrome. Finally, patients who developed inhibitors were followed.

The main outcomes were the development of inhibitors, severity of bleeding, trauma, hospitalizations, and anaphylactic reactions. These outcomes were determined by the information available in the medical records. Additionally, data related to treatments performed in the 6 months prior to admission to the Specialized HCP Coagulopathies Program were taken into account. In those patients who developed inhibitors, a descriptive follow-up against complications and management was carried out from the medical records.

### Statistical analysis

**Descriptive data analysis.** The database was created in Microsoft Excel 2020 for Windows. Analyses were performed using the statistical package SPSS 28.0 (IBM, New York, NY, USA). The information was validated to ensure that the variables had been adequately defined and the data is complete. Dichotomous and categorical variables are described as percentages. For continuous variables, the means, medians, interquartile ranges (IQR) and standard deviations (SD) were estimated. Additionally, the normality of continuous variables was analyzed using distribution graphs and the Kolmogorov–Smirnov test. For the analysis of the time to the presentation of the event, the survival function was developed using the Kaplan–Meier

method. Patients who discontinued the prophylaxis and those who were lost to follow-up were censored.

**Inference analysis.** For the analysis of continuous variables with normal behavior, Student's t test was used, and for variables with nonnormal behavior, the Mann–Whitney test was used. For categorical variables, $X^2$ tests and Fisher's test were performed. A bivariate analysis was performed, identifying possible variables associated with bleeding in patients undergoing substitution therapy, for which comparisons were made with sociodemographic, clinical, and pharmacological variables. A p-value <0.05 was established as statistically significant.

## Results

A total of 207 patients undergoing prophylactic treatment were identified. Of these, 40 already had inhibitors before entering the program, five who were not in the program for at least 6 months, two who did not have a complete medical record, and one who was participating in a clinical trial were excluded. Finally, 159 subjects met the inclusion criteria. They were followed for an average of 5.9±2.3 years and had an average age at admission of 23.6±16.1 years. The majority of patients were classified in the severe hemophilia A group. The patients were distributed mainly in the cities of Bogotá/Cundinamarca (n = 87; 54.7%), Cali/Valle del Cauca (n = 28; 17.6%) and Medellín/Antioquia (n = 10; 6.3%., and the rest (n = 34; 21.4%) were from 11 other departments in Colombia. Fifty-three 53 patients had some family history of hemophilia (Table 1).

In the cohort, 47 patients (29.6%) had some associated comorbidities, the most frequent were arterial hypertension (n = 12; 7.5%), non-ulcer dyspepsia (n = 3; 1.9%), type 2 diabetes mellitus (n = 1; 0.6%) and chronic obstructive pulmonary disease (COPD) (n = 1; 0.6%). However, they also had other infections, such as hepatitis C virus (HCV) infection (n = 24; 15.1%), human immunodeficiency virus (HIV) (n = 2; 1.2%) and hepatitis B virus (HBV) (n = 2; 1.2%). Patients had these infections for more than 10 years on average, in total, at the time of admission to the program and had an average of 39.9 years old.

Prior to admission to the cohort, 121 (76.1%) patients were receiving prophylaxis (Previously Treated Patients–PTPs), and none of these had low or high response inhibitors. Table 1 shows the proportions of patients according to the type of hemophilia, severity, age at the time of diagnosis, family history, complications prior to admission (especially arthropathy) and previous management.

The most frequent type of management was secondary prophylaxis (n = 71; 44.7%), followed by tertiary prophylaxis (Table 2). The most frequently used coagulation factor was recombinant (n = 84, 52.8% of all patients); in 68 patients (80.9%), there was no clinical record of the reason for its use, while in 8 (9.5%) cases, the reported reason was previous hypersensitivity to the plasma derivative. Table 2 also shows the distribution of patients according to the type of prophylaxis, the factors used, the weekly doses of substitution therapy and other reasons for switches or modifications of treatment (in 42 cases, 26.4%) during the follow-up period, in addition to the frequency and causes of discontinuation/interruption of treatment (36 cases, 22.6%), with loss of affiliation with the insurance health company (Entidad Promotora de Salud- EPS) being the most frequent (n = 11; 6.9%).). The median time of discontinuation of treatment was 3 months, mainly due to loss of affiliation and administrative problems with the EPS.

Out of the 159 studied patients; 28 (17.6%) have been switched from one substitution therapy to another., the majority of the switches between the different coagulation factors used was the transition from plasma derivatives to recombinant treatment in both the hemophilia A and hemophilia B patients. Table 3 shows the frequency of switches between the types of coagulation factors used.

**Table 1. Demographic and clinical characteristics of hemophilia A and B patients on prophylaxis prior to follow up; in Colombia, 2012–2019.**

| Characteristics | Frequency (n = 159) | % |
|---|---|---|
| **Hemophilia A** | | |
| Severe | 125 | 89.3 |
| Moderate | 15 | 10.7 |
| **Hemophilia B** | | |
| Severe | 13 | 68.4 |
| Moderate | 6 | 31.6 |
| **History of hemophilia** | | |
| Family history of hemophilia | 53 | 33.3 |
| Family history of inhibitors | 2 | 1.3 |
| **Age at diagnosis** | | |
| 0–12 months | 89 | 56 |
| 12–60 months | 33 | 20.8 |
| > 60 months | 2 | 1.2 |
| No data available | 35 | 22.0 |
| **Previous complications** | | |
| Arthropathy | 126 | 79.2 |
| Number of joints with arthropathy (median, IQR) | 2 (1–4) | |
| Bleeding 6 months prior | 19 | 11.9 |
| Number of bleeds (median, IQR) | 1 (1–1) | |
| **Previous treatments** | | |
| Plasma derived factor IX | 12 | 7.5 |
| Recombinant factor IX | 6 | 3.8 |
| Plasma derived factor VIII | 62 | 39.0 |
| Recombinant factor VIII | 57 | 35.8 |
| No previous treatment | 8 | 5.0 |
| No data available | 14 | 8.8 |
| **Previous weekly factor dose** | | |
| Dose (IU per week. median, IQR) † | 4500 (3000–5000) | |
| Dose (IU/kg per week) (median, IQR) † | 68,2 (56,2–86,9) | |
| On demand | 2 | 1.3 |
| Number applications per week (median, IQR) | 3 (2–3) | |

† Patients with data recorded in clinical records. **IU**: international units. **IQR**: interquartile range

Regarding complications and events in the patients with hemophilia who were receiving prophylaxis, the vast majority of the patients had suffered some type of trauma, almost one-third had undergone some type of surgical intervention, and almost one-fifth had to be hospitalized during the follow-up period. More than 90% had been some type of hemorrhage, most commonly articular and of traumatic origin; of these patients, the majority required the administration of substitution therapy. Only 2 (1.2%) patients had experienced anaphylactic reactions (Table 4).

## Bivariate analysis

**Bleeding.** In determining the association of the variables with the presence of bleeding, statistical significance was found for arterial hypertension (OR:0.162; 95%CI:0.045–0.623;

**Table 2. The frequencies of treatment-related variables among hemophilia A and B patients on prophylaxis in Colombia.**

| Treatment characteristic | Frequency n = 159 | % |
|---|---|---|
| **Type of prophylaxis** | | |
| Primary | 27 | 17.0 |
| Secondary | 71 | 44.7 |
| Tertiary | 61 | 38.4 |
| **Treatments** | | |
| Plasma derived factor IX | 11 | 6.9 |
| Recombinant factor IX | 8 | 5 |
| Plasma derived factor VIII | 62 | 39 |
| Recombinant factor VIII/ von Willebrand factor | 1 | 0.6 |
| Recombinant factor VIII | 77 | 48.4 |
| **Weekly factor dose** | | |
| Dose (IU per week) (median, IQR) † | 4500 (2500–5000) | |
| Dose (IU/kg per week) (median, IQR) † | 75 (55,5–90,3) | |
| Number applications per week (median, IQR) | 3 (2–3) | |
| Initial treatment duration (months) (median, IQR) | 66 (21–88) | |
| Use of recombinant factors at any time during follow-up | 114 | 71.7 |
| **Reason for use of recombinant factor** | | |
| Prior hypersensitivity to plasma derivatives | 8 | 9.5 |
| Poor response to plasma derivatives | 3 | 3.6 |
| Shortage of plasma derivatives | 3 | 3.6 |
| Clinical preference due to age and risk of infection | 1 | 1.2 |
| Patient request | 1 | 1.2 |
| No data available | 68 | 80.9 |
| **Switch of treatment** | | |
| From recombinant factor to plasma derivative | 1 | 0.6 |
| From plasma-derived to recombinant factor | 27 | 17.0 |
| From recombinant factor to recombinant factor | 3 | 1.9 |
| From plasma-derived to plasma derivate | 9 | 5.7 |
| **Discontinuation** | | |
| Discontinuation of prophylaxis (at least one week) | 36 | 22.6 |
| Time without receiving prophylaxis (months) (median; IQR) | 3 (1–7) | |
| **Causes of interruption/discontinuation** | | |
| Loss of EPS affiliation | 11 | 6.9 |
| Administrative problem with EPS | 10 | 6.3 |
| Poor adherence/not attending to follow-ups | 6 | 3.8 |
| Patient's decision | 3 | 1.9 |
| Transfer to area without coverage | 2 | 1.3 |
| Medical decision (improvement in bleeding frequency) | 1 | 0.6 |
| Anaphylactic reaction | 1 | 0.6 |
| No data available | 2 | 1.3 |

† Patients with data recorded in clinical records.

EPS: **Insurance Health Company (Entidad Promotora de Salud)**. IU: **international units**. IQR: **interquartile range**.

**Table 3. The frequencies and types of treatment switching among hemophilia A and B patients on prophylaxis in Colombia.**

| | Total of patients | Non switchers | Switch to recombinant factor IX | Switch to recombinant factor VIII | Switch to plasma derived factor VIII |
|---|---|---|---|---|---|
| | | n (%) | n (%) | n (%) | n (%) |
| **Hemophilia A** | | | | | |
| Plasma derived factor VIII | 62 (44.3%) | 40 (64.5%) | NA | 22 (35.5%) | 0 (0.0) |
| Recombinant factor VIII | 77 (55.0%) | 76 (98.7%) | NA | 0 (0.0) | 1 (0.3%) |
| Recombinant factor VIII/ von Willebrand factor | 1 (0.7%) | 1 (100.0%) | NA | 0 (0.0) | 0 (0.0) |
| **Hemophilia B** | | | | | |
| Plasma derived factor IX | 11 (57.9%) | 6 (54.5%) | 5 (45.5%) | NA | NA |
| Recombinant factor IX | 8 (42.1%) | 8 (100.0%) | 0 (0.0%) | NA | NA |

**NA**: not applicable

p = 0.016), and those from the department of Cauca (OR:0.114; 95%CI:0.023–0.571; p = 0.019) had a lower probability of experiencing hemorrhage.

**Hospitalizations.** None of the examined variables were significantly associated with the need for hospitalization in patients with hemophilia undergoing prophylactic management.

**Development of inhibitors.** The bivariate analysis found that patients with moderate hemophilia had a higher risk of developing inhibitors against coagulation factors (OR:14.421; 95%CI:1.247–166.772; p = 0.046). The three patients who developed inhibitors (two with moderate hemophilia A and one with severe hemophilia A) during the observation period underwent additional follow-up. Two of these patients had low response inhibitors that became negative during the observation period; they did not require switches in the prophylactic medication used, but the dose was increased to 500 IU three times a week (previously 250 IU/three times a week (mean 27.7 IU/kg/dose in those patients). One patient with high response inhibitors (35.9 BU) was identified; this patient was a 2-year-old infant with severe hemophilia A. The patient showed continued immunotolerance of recombinant factor VIII (250 IU 1 time per week); factor VIIa (Novoseven® 1 mg/3 times week) was added, followed in February 2018 by the addition of human plasma protein with activity against factor VIII inhibitors at a dose of 1000 IU 3 times a week. This regimen was continued for 22 months until the patient achieved negative inhibitors in January 2020. A slight increase to 1.1 BU was observed at the last evaluation (April 2020). It should be noted that this same patient presented 21 bleeds due to traumatic causes with factor requirements from the diagnosis of inhibitors to the last reporting date.

## Discussion

Based on the information for the cohort of patients with hemophilia A and B receiving prophylactic management at a IPS-E, it was possible to identify the sociodemographic and clinical characteristics and clinical outcomes of interest, such as the frequency of bleeding, trauma and inhibitor development during follow-up, in addition to the drugs used, their doses and any switches. The evaluation of these results allows the continuous improvement of the health system [11, 12]. Additionally, the information collected could contribute to the development of mechanisms to guide therapeutic strategies, access to pharmacological treatments and the generation of knowledge based on real-world evidence of patients with hemophilia in Colombia.

Within the selected cohort of patients with severe hemophilia, more than half (56%) were diagnosed in the first 12 months of life, and the remaining cases for whom this data were

**Table 4. The frequencies of different complications reported among hemophilia A and B patients during the follow-up period on prophylaxis in Colombia.**

| | Frequency | % | Hemophilia A | % | Hemophilia B | % |
|---|---|---|---|---|---|---|
| **Bleedings** | | | | | | |
| Number of patients | 159 | | 140 | | 19 | |
| Patients with bleed during follow-up | 144 | 90.6 | 128 | 91.4 | 16 | 84.2 |
| Total frequency of patients during follow-up (median; IQR) | 11.3 | 12.2 | 11.4 | 12.6 | 10.6 | 9.1 |
| Total of follow-up (months. median (IQR)) | 7 (5–8) | | 6.8 (4.8–7.7) | | 7.6 (6.9–7.7) | |
| Mean (# patient bleeds / year) | 20.5 | | 18.8 | | 2.1 | |
| *Type of first bleeding* | | | | | | |
| Spontaneous | 31 | 21.5 | 25 | 19.5 | 6 | 37.5 |
| Joint | 15 | 10.4 | 15 | 11.7 | 3 | 18.8 |
| No joint | 16 | 11.1 | 10 | 7.8 | 3 | 18.8 |
| Traumatic | 97 | 67.4 | 88 | 68.8 | 9 | 56.3 |
| Joint | 71 | 49.3 | 66 | 51.6 | 5 | 31.3 |
| No joint | 26 | 18.1 | 22 | 17.2 | 4 | 25.0 |
| Not classified | 16 | 11.1 | 15 | 11.7 | 1 | 6.3 |
| Number of bleeding with factor requirement | 144 | 100 | 128 | 100 | 16 | 100 |
| **Trauma** [a] | | | | | | |
| Number of patients | 137 | 86.2 | 121 | 86.4 | 16 | 84.2 |
| Frequency (median; IQR) | 5 | (2–11) | 5 | (1.2–11.7) | 5 | (2–10) |
| Incidence (100 patients / year) | 143 | | 177.9 | | 21.1 | |
| *Site* | | | | | | |
| Joint | 273 | 74.0 | 246 | 75.9 | 27 | 60.0 |
| Head | 17 | 4.6 | 16 | 4.9 | 1 | 2.2 |
| Muscular | 70 | 19.0 | 54 | 16.7 | 16 | 35.6 |
| ENT | 9 | 2.4 | 8 | 2.5 | 1 | 2.2 |
| **Surgery** | | | | | | |
| Number of patients | 47 | 29.6 | 38 | 27.1 | 9 | 47.4 |
| Frequency (median; IQR) | 0 | (0–1) | 0 | (0–1) | 0 | (0–0) |
| Incidence (100 patients / year) | 5.9 | | 2.5 | | 9.0 | |
| *Type* | | | | | | |
| Major | 24 | 15.1 | 18 | 12.9 | 6 | 31.6 |
| Minor | 32 | 20.1 | 26 | 18.5 | 6 | 31.6 |
| *Mayor frequent surgeries* | | | | | | |
| Hip replacement | 6 | 3.7 | 3 | 2.1 | 3 | 15.8 |
| Knee replacement | 3 | 1.8 | 3 | 2.1 | 0 | 0.0 |
| Knee arthroscopy | 2 | 1.3 | 2 | 1.4 | 0 | 0.0 |
| Other | 13 | 8.2 | 10 | 7.1 | 3 | 15.8 |
| *Minor frequent surgeries* | | | | | | |
| Tooth extraction | 18 | 11.3 | 15 | 10.7 | 3 | 15.8 |
| Skin tumor resection | 4 | 2.5 | 3 | 2.1 | 1 | 5.3 |
| Other | 4 | 2.5 | 2 | 1.4 | 2 | 10.5 |
| **Hospitalizations** | | | | | | |
| Number of patients | 27 | 17 | 25 | 17.9 | 2 | 10.5 |
| Frequency (median; IQR) | 0 | (0–1) | 0 | (0–1) | 0 | (0–1) |
| Incidence (100 patients / year) | 3.7 | | 3.7 | | 0.3 | |
| *Cause* | | | | | | |
| Bruising | 11 | 31.4 | 11 | 33.3 | 0 | 0.0 |
| Hemarthrosis | 5 | 14.3 | 5 | 15.2 | 0 | 0.0 |

*(Continued)*

**Table 4.** (Continued)

| | Frequency | % | Hemophilia A | % | Hemophilia B | % |
|---|---|---|---|---|---|---|
| Other bleedings | 4 | 11.4 | 2 | 6.1 | 2 | 11.8 |
| Other causes | 15 | 42.9 | 15 | 45.5 | 15 | 88.2 |
| **High dose of factor** | | | | | | |
| Hip replacement surgery patient | 1 | 0.6 | 1 | | 0 | 0 |
| **Inhibitors** | | | | | | |
| *Number of patients* | 3 | 1.9 | 3 | 2.1 | 0 | 0 |
| Low response | 2 | 1.3 | 2 | 1.4 | 0 | 0 |
| High response | 1 | 0.6 | 1 | 0.7 | 0 | 0 |
| Incidence (100 patients / year) | 0.3 | | 0.4 | | 0 | |
| Time of inhibitor development (days; mean; SD) | 509 (677) | | 509 (677) | | NA | NA |
| **Anaphylactic reactions or nephrotic syndrome** | | | | | | |
| Anaphylactic reactions plasma derived factor IX | 1 | 0.6 | 1 | 0.7 | 0 | 0 |
| Anaphylactic reactions recombinant factor IX | 1 | 0.6 | 1 | 0.7 | 0 | 0 |
| Nephrotic syndrome | 1 | 0.6 | 1 | 0.7 | 0 | 0 |

**ENT**: Ear-Nose-Throat. **IQR**: interquartile range. **SD**: standard deviation.

[a]: Trauma included any event with and without bleeding after an injury (regardless of the mechanism or severity), with registry in the medical record.

available were diagnosed between 12 and 60 months of age (22%); this finding is consistent with the reports of epidemiological studies in which patients with severe forms of the disease were diagnosed during the first years of life [3, 13]. This behavior may be related to the adoption of a regulatory framework that promotes comprehensive and timely access to orphan diseases [2]. In addition, patients who entered the follow-up cohort during the observation period had an average age of approximately 24 years, which implies a long evolution of the disease and complications prior to admission to the Coagulopathies Program, as is evident in the study. A high proportion of patients with arthropathy at admission (approximately 80%) had two affected joints, a situation to which clinicians responsible for their care and attention should be alert given its impact on the burden of the disease, quality of life and the need for tertiary prevention. Studies conducted in Austria with patients with severe hemophilia show comparable proportions of arthropathy (76.7%) [11], a complication that can be reduced in these patients with adequate prophylaxis, as reported by Manco-Johnson et al. [12] and as recommended by the World Federation of Hemophilia [2].

The standard of treatment for severe cases is the regular replacement of factor through prophylaxis with clotting factor replacement or other hemostatic products to prevent bleeding; it is recommended that such prophylaxis be started at an early age to avoid musculoskeletal complications [4, 14]. However, the present study shows that only 17% of the cohort were receiving primary prophylaxis; this explains the high number of patients who had complications prior to inclusion in the cohort, as many of them were receiving secondary and tertiary prophylaxis [15]. This situation is consistent with studies conducted in the United States showing that between 1999 and 2010, the proportion of patients receiving prophylaxis increased from 40% to 80%, which had an impact on the reduction of new bleeding [12].

In the present study, the frequency of recombinant factor use increased from 36% prior to admission to the cohort to 47.8%, a finding that was associated with hypersensitivity and inadequate response to previous therapies. The most frequent change was from plasma derivatives to recombinants. Similar findings were reported in other studies, such as that of Ay et al., which reported that as of 2020, up to 74.5% of patients with severe forms of the disease were

using recombinants [11] and indicated that 3 administrations per week of factor VIII was the most frequent interval [11, 12].

The interruption or discontinuation of prophylaxis for at least one week in almost a quarter of patients throughout the follow-up period was one of the most relevant problems and was often associated with administrative problems with the EPS and loss of insurance status, a situation that can cause the patient to miss check-up visits and increases the probability of negative outcomes. The adherence report by Tencer et al. showed a median adherence of 76% in the United States [16]; additionally, Khair K et al. in a multicenter study in the United States, Canada, United Kingdom and Sweden, found a median adherence rate of 58.8% [17], and a similar study identified barriers to adherence to prophylaxis, including issues associated with health systems and others related to the patient and his family [18].

According to the patients' clinical records, 90.6% of the patients had at least one bleeding event during the follow-up period. This finding is consistent with reports of other series of patients with hemophilia A and B, which indicated that between 92.0% and 94.2.% of patients experienced a bleeding event from 1997 to 2003 [19]. However, a finding comparable to 9.4% of the patients did not experience bleeding with prophylaxis over an average of almost 6 years of follow-up is comparable to reports from developed countries of rates between 5.7% and 41% [5, 11]. Considering that the majority of the patients in the present study were in secondary and tertiary prophylaxis, the incidence of 19.2 bleeds per 1000 patients per year (1.9 mean bleeds per year per patient) represents a lower mean annual bleeding rate than that reported for hemophilia A study in the United States by Yan S et al.; who reported that between 2.5 and 3.9 bleeds per patient per year [20]. The differences reported between different countries can be attributed to the dose, the type of mutation and the type of clinical follow-up, as well as patient differences in adherence and behavior [3, 11, 21].

In the evaluated group, the low frequency of development of inhibitors of coagulation factors, and even the recognition that most cases reflected a low response that later became negative, can be explained by the appropriate follow-up of the patients in this program. However, additional studies are required to confirm this association, given the low number of patients in this study who developed inhibitors. Another study reported that a history of not knowing the type of factor a patient used before entering a hemophilia treatment cohort may be associated with an increased risk of inhibitor development [22].

Some limitations should be recognized, such as the fact that data were obtained from outpatient clinical records; however, the care process was detailed in the medical records, which allowed the collection of complete and high-quality information. Other limitations include the small number of patients with severe/moderate hemophilia who met the inclusion criteria and therefore the low frequency of some outcomes, which did not allow statistical analysis. Finally, there was significant heterogeneity in the identified patients as a result of the long evolution of the disease prior to admission to the follow-up cohort, which led to a high frequency of complications and a lack of information about the follow-up and treatments that the patients had received in previous years. In addition, strengths of the study are recognized, such as the complete follow-up of each of the patients for a long period and the description of their treatments and outcomes, which resulted in the complete collection and analysis of the data, and the inclusion of a larger sample of patients than has not previously been reported in Colombia.

## Conclusion

These findings show that in this cohort of patients with hemophilia A and B undergoing prophylactic management, especially those undergoing secondary and tertiary, the use of recombinant coagulation factors led to bleeding rates similar to those reported by high-income

countries and a frequency of complications comparable to the reports in the published literature, which are mainly based on the experiences of developed countries.

## Author Contributions

**Conceptualization:** Jorge E. Machado Alba, Juan Manuel Reyes, Natalia Castaño-Gamboa, Manuel E. Machado-Duque.

**Data curation:** Jorge E. Machado Alba, Juan David Wilches-Gutierrez, Harrison David Ospina-Arzuaga, Andres Gaviria-Mendoza, Luis Fernando Valladales-Restrepo, Manuel E. Machado-Duque.

**Formal analysis:** Harrison David Ospina-Arzuaga, Andres Gaviria-Mendoza, Manuel E. Machado-Duque.

**Funding acquisition:** Jorge E. Machado Alba, Juan Manuel Reyes, Natalia Castaño-Gamboa.

**Investigation:** Jorge E. Machado Alba, Juan David Wilches-Gutierrez, Diana Rocio Arias-Osorio, Andres Gaviria-Mendoza, Manuel E. Machado-Duque.

**Methodology:** Jorge E. Machado Alba, Juan David Wilches-Gutierrez, Diana Rocio Arias-Osorio, Juan Manuel Reyes, Maria Lourdes Nakandakari, Andres Gaviria-Mendoza, Luis Fernando Valladales-Restrepo, Manuel E. Machado-Duque.

**Project administration:** Jorge E. Machado Alba.

**Resources:** Jorge E. Machado Alba, Harrison David Ospina-Arzuaga.

**Software:** Manuel E. Machado-Duque.

**Supervision:** Jorge E. Machado Alba, Maria Lourdes Nakandakari.

**Validation:** Juan David Wilches-Gutierrez, Diana Rocio Arias-Osorio, Maria Lourdes Nakandakari.

**Visualization:** Juan Manuel Reyes, Natalia Castaño-Gamboa.

**Writing – original draft:** Andres Gaviria-Mendoza, Luis Fernando Valladales-Restrepo, Manuel E. Machado-Duque.

**Writing – review & editing:** Jorge E. Machado Alba.

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
