## [Decision Letter · Decision Letter 0]

27 Jan 2023

PONE-D-22-11504Frequency of complications in a cohort of patients diagnosed with hemophilia A or hemophilia B receiving prophylactic treatment at a health care institution in Colombia: Retrospective noninterventional studyPLOS ONE

Dear Dr. Machado-Alba,

Thank you for submitting your manuscript to PLOS ONE. After careful consideration, we feel that it has merit but does not fully meet PLOS ONE’s publication criteria as it currently stands. Therefore, we invite you to submit a revised version of the manuscript that addresses the points raised during the review process.

lease submit your revised manuscript by 12/3. If you will need more time than this to complete your revisions, please reply to this message or contact the journal office at plosone@plos.org. Please include the following items when submitting your revised manuscript:A rebuttal letter that responds to each point raised by the academic editor and reviewer(s). You should upload this letter as a separate file labeled 'Response to Reviewers'.A marked-up copy of your manuscript that highlights changes made to the original version. You should upload this as a separate file labeled 'Revised Manuscript with Track Changes'.An unmarked version of your revised paper without tracked changes. You should upload this as a separate file labeled 'Manuscript'.

We look forward to receiving your revised manuscript.

Kind regards,

Roza Chaireti

Academic Editor

PLOS ONE

Journal Requirements:

2. You indicated that you had ethical approval for your study. In your Methods section, please ensure you have also stated whether you obtained consent from parents or guardians of the minors included in the study or whether the research ethics committee or IRB specifically waived the need for their consent."

4. Thank you for providing the following Funding Statement: 

” yes: Reyes JM, Lourdes M and Castano N are paid employees of Pfizer Colombia. Juan David Wilches-Gutierrez, Diana Rocio Arias-Osorio are paid employees of IPS-Especializada. Andres Gaviria-Mendoza, Natalia Castaño-Gamboa, Luis Fernando Valladales-Restrepo, Manuel E. Machado-Duque and Jorge Machado-Alba are paid employees of Audifarma SA. Harrison David Ospina-Arzuaga do not have any conflict of interest.”

We note that one or more of the authors is affiliated with the funding organization, indicating the funder may have had some role in the design, data collection, analysis or preparation of your manuscript for publication; in other words, the funder played an indirect role through the participation of the co-authors.

If the funding organization did not play a role in the study design, data collection and analysis, decision to publish, or preparation of the manuscript and only provided financial support in the form of authors' salaries and/or research materials, please review your statements relating to the author contributions, and ensure you have specifically and accurately indicated the role(s) that these authors had in your study in the Author Contributions section of the online submission form. Please make any necessary amendments directly within this section of the online submission form.  Please also update your Funding Statement to include the following statement: “The funder provided support in the form of salaries for authors [insert relevant initials], but did not have any additional role in the study design, data collection and analysis, decision to publish, or preparation of the manuscript. The specific roles of these authors are articulated in the ‘author contributions’ section.”

If the funding organization did have an additional role, please state and explain that role within your Funding Statement.

Please also provide an updated Competing Interests Statement declaring this commercial affiliation along with any other relevant declarations relating to employment, consultancy, patents, products in development, or marketed products, etc. 

Reviewers' comments:

Reviewer's Responses to Questions

**Comments to the Author**

1. Is the manuscript technically sound, and do the data support the conclusions?

Reviewer #1: Partly

Reviewer #2: Partly

Reviewer #3: Partly

2. Has the statistical analysis been performed appropriately and rigorously? 

Reviewer #1: Yes

Reviewer #2: Yes

Reviewer #3: Yes

3. Have the authors made all data underlying the findings in their manuscript fully available?

Reviewer #1: Yes

Reviewer #2: No

Reviewer #3: Yes

4. Is the manuscript presented in an intelligible fashion and written in standard English?

Reviewer #1: Yes

Reviewer #2: Yes

Reviewer #3: No

5. Review Comments to the Author

Reviewer #1: In their manuscript authors followed a retrospectice cohort, trying to define outcomes of therapy (prophylaxis/ on demand, plasma deriverd/ recombinanat coagulation products) eg: bleeding, inhibitor formation among patients treated between 2012-2019. The manuscript is well written yet showes no novelty and could be improved by additional dAZta collection (eg: mutation type, if available, additional data re emicizumab tretament (approved after 2019). Authors are advised to resubmit into na hemophilia specific journal

Reviewer #2: This manuscript gives an overview a substantial group of patients receiving prophylaxis in Colombia. These real world data give important information about the effectiveness and complications of prophylactic treatment.

I have some suggestions to improve the manuscript.

First of all, for readers not familiair with the Coagulopathies Program of IPS-E (such as myself) it would be helpful to give a short background about te program, who is included, will this lead to selection bias influencing the data?

Minor suggestions:

- I think the proportion of patients receiving plasma-deriverd products is rather high, although lower than 50%. Using the words 'mainly' and 'most frequently' is misleading with such a small difference.

- I would suggest altering the term 'antihemophilic factor' into 'substitution therapy / clotting factor replacement

- bivariate analysis: which variables were investigated?

- Table 1: The column % is not always a percentage

- Table 1 and 2: dose (IU per week) would be more informative if changed into dose/kg/wk

- Please provide data about the number of exposure days before inhibitor develpment occurred

- in the patients with the anaphylactidc reaction, both haemophiliia B patients, was this associated with inhibitor development?

- Table 2: under 'weekly factor dose', what dose ;use of recombinant factors' mean?

- Table 3: 'total of follow up ' what does this mean?

- Table 3: why are patients with bruising admitted to the hospital?

Reviewer #3: General comments

- The topic is important, however, the paper needs revision and further editing.

- All comments are mentioned on the text of the paper.

Specific comments

Title: the suggested title is " The Frequency of complications in a cohort of patients with hemophilia A and B receiving prophylactic treatment in Colombia: a retrospective non-interventional study".

Introduction:

- Line 72; Hemophilia A and B are rare diseases worldwide and not only in Colombia.

- Line 99: Provider needs to be replaced by Institution

Materials and methods:

- Line 110: Does this statement mean that authors stop following patients when they develop inhibitors? If Yes, Why they did not follow them?

- Line 133: Authors need to clarify what they mean by trauma as high percentage of patients had trauma (68%, Table 4).

Results:

- Line 173: the statement needs to be changed as mentioned in the comments.

- The titles of most Tables need to be changed (the suggested titles are mentioned on the manuscript).

- Table 4: This table needs to be reviewed and many details can be mentioned in the text instead, e.g. types of bleeding, trauma and surgeries.

- Line 231: Bivariate analysis; This is very important, it needs to be put in a Table, and only the important findings can be written in the text

Discussion:

- Line 312: Reference 16 needs to be added.

6. PLOS authors have the option to publish the peer review history of their article (what does this mean?). If published, this will include your full peer review and any attached files.

Reviewer #1: No

Reviewer #2: **Yes: **L.F.D. van Vulpen

Reviewer #3: **Yes: **Meaad Kadhum Hassan

---

## [Author Response · Author response to Decision Letter 0]

21 Feb 2023

Reference ID: PONE-D-22-11504

Title: The frequency of complications in a cohort of patients diagnosed with hemophilia A and hemophilia B receiving prophylactic treatment in Colombia: a retrospective noninterventional study

Dear editors

PLoS One

We responded to each of the comments made by the editor.

Declarations 

Declaration of interests: Reyes JM, Lourdes M and Castaño-Gamboa N are paid employees of Pfizer Colombia; this does not alter our adherence to PLOS ONE policies on sharing data and materials. Juan David Wilches-Gutierrez, Diana Rocio Arias-Osorio are paid employees of IPS-Especializada. Andres Gaviria-Mendoza, , Luis Fernando Valladales-Restrepo, Manuel E. Machado-Duque and Jorge Machado-Alba are paid employees of Audifarma SA. Harrison David Ospina-Arzuaga do not have any conflict of interest. No other relevant declarations relating to employment, consultancy, patents, products in development, or marketed products.

Funding:

This study was funded by Pfizer Colombia. The funders had role in study design and decision to publish

Author contributions: JEMA participated in the study design, drafting, data collection, data analysis, description of results, discussion, critical revision of the article, and evaluation of the final version of the manuscript. LFVR, AGM, MEMD participated in the study design, drafting, data collection, data analysis, description of results and discussion. HDOA participated in collect information and results. JDWG and DRAO participated in the study design, drafting, and discussion of the article. JMR, NCG and MLN participated in study design and discussion.

---

## [Decision Letter · Decision Letter 1]

11 May 2023

The frequency of complications in a cohort of patients diagnosed with hemophilia A and hemophilia B receiving prophylactic treatment in Colombia: a retrospective noninterventional study

PONE-D-22-11504R1

Dear Dr. Machado-Alba,

We’re pleased to inform you that your manuscript has been judged scientifically suitable for publication and will be formally accepted for publication once it meets all outstanding technical requirements. Please note that one of the reviewers has two minor comments (refer to attached file) - you can make the changes when you check the draft.

Kind regards,

Roza Chaireti

Academic Editor

PLOS ONE

Additional Editor Comments (optional):

Reviewers' comments:

Reviewer's Responses to Questions

**Comments to the Author**

1. If the authors have adequately addressed your comments raised in a previous round of review and you feel that this manuscript is now acceptable for publication, you may indicate that here to bypass the “Comments to the Author” section, enter your conflict of interest statement in the “Confidential to Editor” section, and submit your "Accept" recommendation.

Reviewer #2: All comments have been addressed

Reviewer #3: All comments have been addressed

2. Is the manuscript technically sound, and do the data support the conclusions?

Reviewer #2: Yes

Reviewer #3: Yes

3. Has the statistical analysis been performed appropriately and rigorously? 

Reviewer #2: I Don't Know

Reviewer #3: Yes

4. Have the authors made all data underlying the findings in their manuscript fully available?

Reviewer #2: Yes

Reviewer #3: Yes

5. Is the manuscript presented in an intelligible fashion and written in standard English?

Reviewer #2: Yes

Reviewer #3: Yes

6. Review Comments to the Author

Reviewer #2: No furter comments, previous comments are adequatly adressed. This manuscript gives a nice overview of the haemophilia patients receiving expensive treatment in a low resource country.

Reviewer #3: Most of the comments were done by the authors. Only 2 comments (in the attached file. minor) need to be corrected.

These are present on the attached file.

Regards

7. PLOS authors have the option to publish the peer review history of their article (what does this mean?). If published, this will include your full peer review and any attached files.

Reviewer #2: No

Reviewer #3: No

---

## [Editor Report · Acceptance letter]

15 May 2023

PONE-D-22-11504R1 

The frequency of complications in a cohort of patients diagnosed with hemophilia A and hemophilia B receiving prophylactic treatment in Colombia: a retrospective noninterventional study 

Dear Dr. Machado Alba:

I'm pleased to inform you that your manuscript has been deemed suitable for publication in PLOS ONE. Congratulations! Your manuscript is now with our production department. 

Kind regards, 

on behalf of

Dr. Roza Chaireti 

Academic Editor

PLOS ONE